# Extracellular Cystatin F Is Internalised by Cytotoxic T Lymphocytes and Decreases Their Cytotoxicity

**DOI:** 10.3390/cancers12123660

**Published:** 2020-12-06

**Authors:** Mateja Prunk, Milica Perišić Nanut, Tanja Jakoš, Jerica Sabotič, Urban Švajger, Janko Kos

**Affiliations:** 1Department of Biotechnology, Jožef Stefan Institute, Jamova cesta 39, 1000 Ljubljana, Slovenia; mateja.prunk@ijs.si (M.P.); milica.perisic@ijs.si (M.P.N.); jerica.sabotic@ijs.si (J.S.); 2Faculty of Pharmacy, University of Ljubljana, Aškerčeva cesta 7, 1000 Ljubljana, Slovenia; tanja.jakos@ffa.uni-lj.si; 3Blood Transfusion Centre of Slovenia, Šlajmerjeva 6, 1000 Ljubljana, Slovenia; urban.svajger@ztm.si

**Keywords:** cystatin F, cathepsins, granzymes, perforin, TALL-104, cytotoxic lymphocytes

## Abstract

**Simple Summary:**

Cytotoxic T lymphocytes kill cancer or virally infected cells by exocytosis of lytic granules. This leads to perforin-mediated granzyme entry into the target cell, consequently killing the target cell. Granzymes and perforin are activated by cysteine cathepsins whose activity is regulated by the protein inhibitor cystatin F. Since cystatin F can be secreted by a range of cancer and immune cells in tumour microenvironments, we here investigated whether extracellular cystatin F can be taken up by and affect the function of cytotoxic T lymphocytes. We demonstrated cystatin F uptake into cytotoxic T lymphocytes, down-regulation of target peptidases, and reduced target cell killing. Overall, our results indicate that cystatin F is an important mediator that can impair the killing efficiency of cytotoxic T lymphocytes and thus suggest that it is a possible target for cancer immunotherapy.

**Abstract:**

Cystatin F is a protein inhibitor of cysteine cathepsins, peptidases involved in the activation of the effector molecules of the perforin/granzyme pathway. Cystatin F was previously shown to regulate natural killer cell cytotoxicity. Here, we show that extracellular cystatin F has a role in regulating the killing efficiency of cytotoxic T lymphocytes (CTLs). Extracellular cystatin F was internalised into TALL-104 cells, a cytotoxic T cell line, and decreased their cathepsin C and H activity. Correspondingly, granzyme A and B activity was also decreased and, most importantly, the killing efficiency of TALL-104 cells as well as primary human CTLs was reduced. The N-terminally truncated form of cystatin F, which can directly inhibit cathepsin C (unlike the full-length form), was more effective than the full-length inhibitor. Furthermore, cystatin F decreased cathepsin L activity, which, however, did not affect perforin processing. Cystatin F derived from K-562 target cells could also decrease the cytotoxicity of TALL-104 cells. These results clearly show that, by inhibiting cysteine cathepsin proteolytic activity, extracellular cystatin F can decrease the cytotoxicity of CTLs and thus compromise their function.

## 1. Introduction

Cytotoxic T lymphocytes CD8+ (CTLs) and natural killer (NK) cells are crucial players in the immune response against virally infected cells and tumour cells [1]. The main target cell killing mechanism they use in this context is lytic granule exocytosis, which releases, most notably, perforin and granzymes. Granzymes are serine peptidases that trigger cell death upon entering the cell via the pore-forming protein perforin [1,2]. Both perforin and granzymes are synthesised as inactive precursors, which are proteolytically activated by cysteine cathepsins [3,4,5].

In cytotoxic lymphocytes, cathepsin C is the major progranzyme convertase, activating progranzymes by removing a dipeptide at their N-terminal end [4]. In the absence of cathepsin C, cathepsin H can act as an additional progranzyme convertase, at least for granzyme B [5]. Cysteine cathepsins also activate perforin precursors by cleaving their C-terminal ends [3,6], and cathepsin L has been suggested to play a major role in this process [6,7].

In immune cells, cystatin F, an endogenous protein inhibitor, was described to play an important role in regulating cysteine cathepsin activity [8]. Cystatin F belongs to type II cystatins; however, unlike other members that are mainly secreted, cystatin F is also localised intracellularly in lysosomal/endosomal compartments [9]. In addition, its inhibitory activity is tightly regulated as, after synthesis, it forms disulphide-linked dimers that do not inhibit cysteine cathepsins [10]. To be active inhibitors of cysteine cathepsins, cystatin F dimers must monomerise. This requires strong reducing conditions [10]; however, the process is highly facilitated by the truncation of the N-terminal end [11]. In addition, N-terminal truncation significantly changes the inhibitory profile of cystatin F. Whereas the full-length monomeric form does not inhibit cathepsin C, N-terminally truncated cystatin F is a potent inhibitor of cathepsin C with an equilibrium inhibitory constant in the nanomolar range [11]. Moreover, cystatin F can be secreted and internalised by bystander cells [12] in which its intracellular localisation and uptake is driven by its glycosylation [12,13,14].

In cytotoxic cells, cystatin F is localised in cytotoxic granules in which it co-localises with its target cathepsins C, H, and L [11,15,16] indicating that cystatin F regulates their activities and thus the activation of the effector molecules of the perforin/granzyme pathway [8,17]. Indeed, it was demonstrated that extracellular cystatin F decreases the activities of granzymes A and B as well as the cytotoxicity of NK-92 and primary human NK cells [13]. In addition, both dimeric and monomeric cystatin F levels were increased in split anergic primary NK cells [15], a state characterised by decreased cytotoxicity and increased cytokine secretion [18,19]. Similarly, cystatin F levels were increased while the activities of cathepsins C, H, and L and granzyme B were decreased in the cytotoxic T CD8+ cell line (TALL-104) when a hyporesponsive anergic state was induced by ionomycin or transforming growth factor beta [16]. Likewise, the overexpression of cystatin F in mouse CTLs was shown to attenuate cathepsin C activity [11], and internalisation of cystatin F secreted from mouse CTLs into cystatin F-null mouse CTLs was also demonstrated [14]. However, whether extracellular cystatin F can directly affect cysteine cathepsin activity in recipient CTLs and decrease their cytotoxicity has not been demonstrated yet.

In this study, we demonstrate that extracellular cystatin F decreases CTL cytotoxicity, likely by affecting the activation of granzymes, effector molecules of the perforin/granzyme pathway. We first demonstrate internalisation of cystatin F into TALL-104 cells and interaction of internalised cystatin F with cathepsin C. We show that extracellular cystatin F attenuates the activities of cathepsins C, H, and L and granzymes A and B, while it does not affect perforin processing. Finally, we demonstrate that cystatin F secreted from target cells can also affect the cytotoxicity of CTLs.

## 2. Results

### 2.1. Extracellular Cystatin F Is Internalised into TALL-104 Cells in Which It Interacts with Cathepsin C and Localises to Cytotoxic Granules

We first assessed recombinant cystatin F uptake into TALL-104 cells, a model cell line of cytotoxic T lymphocytes [16]. Western blot analysis demonstrated that his-tag-labelled recombinant cystatin F is indeed internalised into TALL-104 cells (Figure 1a,b). Recombinant cystatin F uptake was further confirmed by confocal immunofluorescence microscopy (Figure 1c). Cystatin F is an inhibitor of cathepsin C, a major progranzyme convertase in cytotoxic lymphocytes, and the interaction between internalised cystatin F and cathepsin C was confirmed by co-immunoprecipitation using an antibody directed against the his-tag (Figure 1d). Finally, by using confocal immunofluorescence microscopy, we demonstrated that cystatin F is at least partially targeted to cytotoxic granules, as his-tagged cystatin F co-localised with perforin and granzyme A (Figure 1e,f).

### 2.2. Internalised Extracellular Cystatin F Decreases Cathepsin C, H, and L Activity

In cytotoxic lymphocytes, cystatin F regulates the activities of cathepsins C, H, and L, which are implicated in granzyme and perforin activation [7,13]. The activities of cathepsins C, H, and L were significantly decreased in TALL-104 cells after incubation with either wild-type or N-terminally truncated forms of cystatin F (Figure 2a–c).

### 2.3. Cystatin F Decreases Granzyme A and B Activity and Cell Cytotoxicity

Next, we aimed to determine whether lower cathepsin C and H activity translates to decreased granzyme activity. Granzyme A and B activity was determined in whole-cell lysates of TALL-104 cells treated with full-length or N-terminally truncated cystatin F. Extracellularly added full-length cystatin F significantly inhibited the activity of granzymes A and B (Figure 2d,e) in TALL-104 cells, and this inhibition was even more pronounced for N-terminally truncated cystatin F.

Next, we tested whether treatment of TALL-104 cells with full-length or N-terminally truncated cystatin F affects their cytotoxic potential by calcein-AM release assay. The ability of TALL-104 cells to kill target K-562 cells was significantly reduced after treatment with either form of cystatin F (Figure 2f,g).

Lastly, to further extend our findings in a more physiological setting we tested the effect of full-length cystatin F on granzyme activity and cytotoxicity of primary human SEA/SEB-stimulated CTLs. Indeed, activities of granzymes A and B were significantly decreased (Figure 3a,b). Importantly, the ability of SEA/SEB-stimulated CTLs to kill SEA/SEB-pulsed Raji cells was also decreased (Figure 3c,d).

### 2.4. Cystatin F Does Not Affect Pro-Cathepsin C and Pro-Perforin Processing in TALL-104 Cells

Since cystatin F is a potent inhibitor of cathepsin L, which is involved in the maturation of cathepsin C [20], we next assessed whether cystatin F affects pro-cathepsin C processing and whether the observed decrease in cathepsin C activity (Figure 2a) is due to direct cathepsin C inhibition by extracellular cystatin F or is a consequence of inhibited maturation. First, we transfected HeLa cells that do not express cystatin F with full-length cystatin F and analysed the levels of precursor and active forms of cathepsin C. After transfection with cystatin F, the levels of mature cathepsin C decreased, while the levels of the precursor form increased (Figure 4a–c), revealing that cystatin F can inhibit cathepsin C maturation. However, when we analysed cathepsin C levels in TALL-104 cells treated with either full-length or N-terminally truncated cystatin F, no difference was observed between the levels of mature and precursor forms (Figure 4d). In addition, cystatin F directly impaired cathepsin H and L activity but had no impact on their protein levels (Figure 4d). Similarly, internalised cystatin F had no effect on perforin processing (Figure 4e,f), neither after shorter (4 h) or prolonged incubation periods (24 h (Figure 4e,f) and 4 days [21]). However, incubation with 20 μM of E64d, a general cysteine cathepsin inhibitor that is known to inhibit perforin processing [3,6], efficiently inhibited perforin processing in TALL-104 cells after 17 h (Figure 4e,f).

### 2.5. Internalisation of Extracellular Cystatin F into Target K-562 Cells Enhances Their Resistance to TALL-104 Cells

Given the effect of cystatin F on TALL-104 cytotoxicity, we assessed whether the presence of cystatin F in the target cells themselves inhibits the cytotoxic effect of lymphocytes, thus conferring protection to target cells. Target K-562 cells that do not express endogenous cystatin F were incubated with recombinant cystatin F. Western blot analysis revealed that full-length as well as N-terminally truncated cystatin F were both internalised into target K-562 cells within 4 h (Figure 5). When cystatin F-pretreated K-562 cells were resuspended in fresh medium without cystatin F for an additional 4 h, the internalised cystatin F was almost completely secreted into the cell medium (Figure 5). Importantly, the calcein-AM release assay revealed that cystatin F-pretreated K-562 target cells were killed less efficiently than untreated K-562 target cells (Figure 6).

## 3. Discussion

Cystatin F is a protein inhibitor of cysteine cathepsins that is found intracellularly in the lysosomal/endosomal pathway as well as extracellularly [9,13]. The extracellular cystatin F is mainly found in the inactive disulphide-linked dimeric form, which can be internalised into bystander cells mainly via the mannose-6-phosphate receptor system [12,14]. After internalisation, cystatin F is targeted to endosomal/lysosomal compartments, in which it can be activated by the proteolytic cleavage of 15 residues at its N-terminal end, enabling its activating monomerisation and regulation of cysteine cathepsin activity in recipient cells [11,12,13,14]. Cysteine cathepsins are involved in the activation of effector molecules of the perforin/granzyme pathway in NK cells and CTLs and since cystatin F is found in the same cellular compartment as its target cysteine cathepsins, it was proposed to play an important role in regulating the effector functions of cytotoxic lymphocytes [22]. Indeed, intracellular cystatin F acts as an upstream regulator of split anergy in NK cells, a functional phenotype that appears after the interaction of NK cells with tumour cells or monocytes and is characterised by decreased cytotoxicity and increased cytokine secretion [15]. Moreover, extracellular cystatin F present in the tumour microenvironment may attenuate granzyme A and B activity and decrease recipient NK cell cytotoxicity [13]. However, whether extracellular cystatin F after internalisation affects cysteine cathepsins in recipient CTLs and, importantly, affects the ability of CTLs to kill their target cells was not demonstrated. Here, we provide evidence that extracellular cystatin F can be fully internalised into cytotoxic TALL-104 cells, and, as in NK cells, can decrease their cytotoxicity by inhibiting granzyme A and B activity (Figure 7). A similar result was obtained using primary human CTLs. Furthermore, we show that cystatin F secreted from target cells has a protective role against the cytotoxic action of TALL-104 cells.

Extracellular cystatin F was demonstrated to be internalised by several cell types, including human HeLa and HEK-293 cell lines [13,14] and murine fibroblast cell line L929 [14], murine bone marrow-derived macrophages [12], bone marrow-derived dendritic cells from cystatin F knock-out mice [12], and CTLs and splenocytes from cystatin F knock-out mice [14]. All of these studies demonstrated uptake by cells that do not express endogenous cystatin F. In NK cells that express endogenous cystatin F, extracellular cystatin F uptake was demonstrated to potentiate the suppressive function of endogenous cystatin F regarding granzyme function and cell cytotoxicity [13]. As the internalised cystatin F in cystatin F-expressing cells must compete with endogenous intracellular cystatin F, the target peptidases might differ between internalised and endogenous cystatin F. However, here we demonstrate that the target of internalised extracellular cystatin F in TALL-104 cells is also cathepsin C, similar to endogenous cystatin F that also forms immune complexes with cathepsin C [11,16].

To confirm the effect of internalised cystatin F on target cathepsins, we analysed cathepsin C, H, and L activity in TALL-104 cells and found that they were significantly reduced. We also compared the effect between full-length and N-terminally truncated cystatin F on their activities. N-terminally truncated cystatin F is a monomeric form and can immediately interact with cathepsins after internalisation. Conversely, full-length cystatin F forms inactive dimers that must first be processed at the N-terminus and monomerise to become active inhibitors [10,11]. Both forms, full-length and N-terminally truncated, showed similar inhibitory activity, suggesting efficient activation of the full-length cystatin F upon internalisation (Appendix A). Besides direct binding and inhibition, cystatin F might decrease cathepsin C activity via inhibiting cathepsin L, which is involved in processing of pro-cathepsin C to its mature form. In HeLa cells that do not express endogenous cystatin F, transfection with full-length cystatin F resulted in lower cathepsin C processing. Nevertheless, evaluating the levels of precursor and mature forms of cathepsin C in TALL-104 cells after incubation with either full-length or N-terminally truncated cystatin F revealed no effect on cathepsin C processing. It is possible that in TALL-104 cells another peptidase, which is not inhibited by cystatin F, activates cathepsin C [20,23]. Another explanation for the observed difference in cathepsin C maturation between HeLa and TALL-104 cells is that HeLa cells were transfected with cystatin F while TALL-104 cells were pretreated with cystatin F; it is likely that internalised cystatin F is solely targeted to cellular compartments in which cathepsin C is already found in the activated form and could therefore directly inhibit cathepsin C and not affect its maturation. Furthermore, we tested the effect of cystatin F on perforin processing through inhibition of cathepsin L [7]. However, in TALL-104 cells, full-length or N-terminally truncated cystatin F did not affect perforin processing. Other peptidases probably compensate decreased cathepsin L activity, as substantial redundancy among peptidases regarding perforin activation has been demonstrated [6]. In contrast to perforin, full-length cystatin F caused a marked decrease in granzyme A and B activity in TALL-104 cells and primary human CTLs. Furthermore, the effect on granzyme A and B activity of N-terminally truncated cystatin F was significantly more pronounced than that of the full-length form in TALL-104 cells. This decrease in granzyme activity had functional consequences, i.e., decreased TALL-104 cell cytotoxicity against K-562 target cells and decreased cytotoxicity of SEA/SEB-activated primary human CTLs against SEA/SEB-pulsed Raji target cells. Treatment with both full-length and N-terminally truncated cystatin F forms led to a significant decrease in TALL-104 cell cytotoxicity. This is consistent with the results obtained for both NK-92 cell line and primary NK cells [13]. However, the difference in the effect on TALL-104 cell cytotoxicity of N-terminally truncated cystatin F compared to wild-type was significantly less pronounced compared to the difference observed in NK-92 cell line and primary NK cells [13]. A possible explanation is that TALL-104 cells might be more efficient in cystatin F activation due to, higher levels of cystatin F-activating peptidase.

Finally, we investigated whether cystatin F in target cells could confer protection against CTL killing. Indeed, cystatin F-pretreated target K-562 cells were killed less efficiently by TALL-104 cells. In the timeframe of the cytotoxicity assay, cystatin F-loaded target K-562 cells released a large portion of internalised cystatin F. Therefore, we propose that target cells may protect themselves from CTL killing by secreting cystatin F, which is subsequently internalised into cytotoxic cells, in which it inhibits the activation of granzymes that are crucial for efficient target cell killing. A protective role of cystatin F in target cells might be especially important in tumour microenvironments in which cystatin F could be exploited by tumour cells to evade the anti-tumour immune response. Indeed, up-regulated cystatin F expression in tumour tissue was demonstrated in patients with colorectal cancer, and was even correlated with liver metastasis and worse prognoses [24]. A similar molecular mechanism was demonstrated for serpinB9, an inhibitor that can directly inhibit granzyme B; tumour cells were shown to protect themselves from cytotoxic lymphocytes by up-regulating serpinB9 [25]. Tumour cells could also internalise cystatin F present in tumour microenvironments into which it could be secreted by several other cell types, especially immune cells such as dendritic cells, monocytes or even cytotoxic lymphocytes [13]. To conclude, our results suggest that cystatin F is an important molecular mediator in tumour microenvironments and is responsible for shaping the anti-tumour immune response and function of cytotoxic lymphocytes.

## 4. Materials and Methods

### 4.1. Antibodies

For western blotting and immunofluorescence microscopy, the rabbit anti-cystatin F antibody from Sigma-Aldrich (St. Louis, MO, USA) and Davids Biotechnologie GmbH (Regensburg, Germany) was used, respectively. Additionally, the following antibodies were used: rabbit anti-β-actin (Sigma-Aldrich), rabbit and mouse anti-glyceraldehyde 3-phosphate dehydrogenase (GAPDH; Proteintech, Rosemont, IL, USA), rabbit anti-his-tag (Cell Signaling Technology, Danvers, MA, USA), rabbit anti-N-terminal part of cystatin F (Amsbio, Abingdon, UK), mouse anti-cathepsin C, mouse anti-granzyme A, and mouse anti-perforin (all from Santa Cruz Biotechnology, Dallas, TX, USA). The mouse anti-cathepsin H antibody 1D10 [26] and sheep anti-cathepsin L antibody [27] were prepared as described previously. For immunofluorescence, the secondary antibodies goat anti-rabbit Alexa Fluor 488 and goat anti-mouse Alexa Fluor 555 (Cell Signaling Technology) were used. For western blot, the following secondary antibodies were used: anti-rabbit, anti-mouse, and anti-sheep secondary antibodies conjugated with horseradish peroxidase (HRP) (Jackson Immuno Research, West Grove, PA, USA); anti-rabbit and anti-mouse secondary antibodies conjugated with fluorescent dyes DyLight 650 and DyLight 550, respectively (Invitrogen, Carlsbad, CA, USA); and anti-mouse secondary antibody conjugated with fluorescent dye StarBright 700 (Bio-Rad, Hercules, CA, USA).

### 4.2. Human Recombinant Cystatin F

Full-length and N-terminally truncated human recombinant cystatin F were produced in the human freestyle HEK293F cell line and purified as described in [13].

### 4.3. Cell Cultures

TALL-104 cells (CRL-11386; ATCC, Manassas, VA, USA) were cultured in Iscove’s Modified Dulbecco’s Medium (IMDM; ATCC) with 20% foetal bovine serum (Gibco, Carlsbad, CA, USA), 100 IU/mL interleukin-2 (Bachem, Dubendorf, Switzerland), 2.5 µg/mL recombinant human albumin (Sigma-Aldrich), and 0.5 µg/mL D-mannitol (Sigma-Aldrich). HeLa cells (CCL-2; ATCC) were cultured in Dulbecco’s Modified Eagle Medium (DMEM; Gibco) with 10% foetal bovine serum (Gibco), 100 U/mL penicillin (Lonza, Basel, Switzerland), and 100 U/mL streptomycin (Lonza). K-562 cells (CCL-243; ATCC) and Raji cells (CCL-86; ATCC) were cultured in RPMI-1640 (Lonza) with 10% foetal bovine serum, 100 U/mL penicillin (Lonza), and 100 U/mL streptomycin (Lonza). Primary human CTLs were isolated from peripheral blood mononuclear cells (PBMC) obtained from healthy volunteers at the Blood Transfusion Centre of Slovenia, Republic of Slovenia, according to institutional guidelines. PBMCs were stimulated with 5 µg/mL staphylococcal enterotoxin A (SEA, Sigma-Aldrich) and 5 µg/mL staphylococcal enterotoxin B (SEB, Sigma-Aldrich) at a cell density of 10^8^/mL for 1 h and then re-suspended in RPMI (Lonza) with 10% foetal bovine serum (Gibco) and 100 IU/mL interleukin-2 (Bachem) at a cell density of 2 × 10^6^/mL. After 5 days SEA/SEB-specific CTLs were isolated by positive selection according to manufacturer’s instructions (Miltenyi Biotech, Bergisch Gladbach, Germany). The purity of CTLs was determined by flow cytometry using fluorescence labelled antibodies against CD3, CD4, CD8, CD56, CD16, CD19 and TCRα/β (all from Miltenyi Biotech) and was >95%.

### 4.4. Confocal Immunofluorescence Microscopy

Cells, untreated or incubated for 4 h with 100 nM full-length cystatin F, were washed once in phosphate-buffered saline (PBS), left to adhere to slides for 30 min at 37 °C, and fixed with 4% paraformaldehyde in PBS for 15 min at room temperature (RT). After rinsing three times with PBS for 5 min each, specimens were blocked for 1 h in 5% goat serum and 0.3% Triton X-100 in PBS. Primary antibodies were diluted in 1% bovine serum albumin and 0.3% Triton X-100 in PBS to a final concentration of 10 μg/mL for anti-cystatin F, 2 μg/mL for anti-his-tag and 4 μg/mL for anti-perforin and anti-granzyme A antibodies and incubated overnight at 4 °C. Next, slides were rinsed three times in PBS for 5 min each and incubated with goat anti-rabbit Alexa Fluor 488 and goat anti-mouse Alexa Fluor 555, diluted 1:1000 in the same buffer as primary antibodies, for 2 h at RT. After the final washing step, the slides were cover-slipped using Prolong Gold Antifade Mountant (Thermo Fisher Scientific, Waltham, MA, USA). Samples without one or both primary antibodies were used as controls. Images were acquired with a Carl Zeiss LSM 710 confocal microscope (Carl Zeiss, Oberkochen, Germany) with ZEN 2011 imaging software (Carl Zeiss).

### 4.5. Preparation of Cell Lysates

Cells were either untreated or incubated with 100 nM full-length or N-terminally truncated cystatin F for 4 h and 24 h or with 20 μM E64d for 17 h. Cells were then washed in PBS, lysed in lysis buffer, incubated for 30 min on ice, and centrifuged at 16,000× *g* for 20 min at 4 °C. Supernatants were transferred to fresh tubes, and protein concentrations were determined using the DC-Protein Assay Kit (Bio-Rad). The lysis buffer for western blot analysis comprised 50 mM Tris-HCl pH 8, 150 mM NaCl, 1% Triton X-100, 0.5% sodium deoxycholate, 0.1% SDS, and 1 mM EDTA with added protease inhibitors (Roche). The lysis buffer for cathepsin activities comprised 0.1 M citrate buffer pH 6.2 with 1% Triton X-100. The lysis buffer for granzyme activities comprised 25 mM HEPES, 250 mM NaCl, 2.5 mM EDTA, and 0.1% Nonidet p-40, pH 7.4.

### 4.6. Western Blot

Samples containing 30–50 µg of cell lysate total protein were resolved in SDS-PAGE using 12% polyacrylamide gel and transferred onto nitrocellulose membranes using the trans-blot turbo transfer system (Bio-Rad) and trans-blot turbo RTA transfer kits with nitrocellulose membrane (Bio-Rad). For some experiments, samples were resolved using 12% TGX stain-free polyacrylamide gels (Bio-Rad); these were activated before the transfer with UV light for 1 min using the ChemiDoc MP System (Bio-Rad). After the transfer, the membranes were imaged for stain-free labelling of total proteins with the ChemiDoc MP System. Next, membranes were blocked in 5% non-fat dry milk in PBS (for cystatin F), 5% non-fat dry milk in tris-buffered saline with 0.1% Tween-20 (for his-tag, N-terminal part of cystatin F and cathepsins C, H, and L), and 1.5% non-fat dry milk/0.5% bovine serum albumin in tris-buffered saline with 0.1% Tween-20 (for perforin). The primary antibodies were diluted in blocking solution at 1:500 for cystatin F and N-terminal part of cystatin F, 1:200 for cathepsin C and perforin, and 1:600 for his-tag and cathepsins H and L and incubated overnight at 4 °C. After washing, the membranes were incubated with HRP- or fluorescently conjugated secondary antibodies diluted 1:5000 in blocking solution. Membranes with HRP-conjugated antibodies were incubated with Lumi-Light western blotting substrate (Roche). Images were acquired using a ChemiDoc MP System (Bio-Rad), and quantification analysis was performed in Image Lab, version 5.1 software (Bio-Rad).

### 4.7. Immunoprecipitation

TALL-104 cells (50 × 10^6^) were either left untreated or were incubated for 4 h with 100 nM recombinant full-length cystatin F with his-tag. Afterwards, the cells were washed with PBS and lysed in lysis buffer (50 mM Tris-HCl pH 7.4, 100 mM NaCl, and 0.25% Triton X) with protease inhibitors. After incubation on ice for 30 min, the cell lysates were centrifuged for 20 min at 16,000× *g*, and the supernatant was transferred to a new tube. The anti-his-tag antibody was added to the cell lysate at a final concentration of 2.7 µg/mL, and the samples were incubated at 4 °C overnight with constant shaking. Dynabeads protein A (Thermo Fisher Scientific) were washed once with lysis buffer, added to the cell lysate, and incubated at RT for 20 min with constant shaking. The beads were then washed three times in lysis buffer, resuspended in SDS loading buffer containing 40 mM dithiothreitol, and boiled for 5 min. The co-immunprecipitated proteins were analysed by western blot.

### 4.8. Determination of Enzyme Activities

Enzyme activities were determined using the following substrates: 70 µM H-Gly-Phe-7-amino-4-methylcoumarin (AMC) (Bachem) for cathepsin C, 20 µM H-Arg-AMC (Bachem) for cathepsin H, 50 µM Z-Phe-Arg-AMC for cathepsin L (Bachem), 50 µM acetyl-Ile-Glu-Pro-Asp-AMC for granzyme B (Bachem), and 200 µM Z-Gly-Pro-Arg-AMC for granzyme A (Bachem). The following assay buffers were used: 25 mM MES, 100 mM NaCl, 5 mM cysteine, pH 6 (for cathepsin C); 100 mM MES, 2 mM EDTA, 5 mM cysteine, pH 6.5 (for cathepsins H and L); 50 mM Tris-HCl, 100 mM NaCl, pH 7.4 (for granzyme B); and 20 mM Tris, 150 mM NaCl, pH 8.1 (for granzyme A). Whole-cell lysates were first activated in assay buffer for 15 min at RT for cathepsins or for 30 min at 37 °C for granzymes. The substrate was then added, and formation of fluorescent products was measured continuously on a microplate reader Infinite M1000 (Tecan, Männedorf, Switzerland). To determine cathepsin L activity, 5 µM of irreversible inhibitor of cathepsin B, CA-074 (Bachem), was added before the addition of substrate.

### 4.9. Transfection

HeLa cells were grown in 6-well plates and transfected with the pcDNA3 plasmid containing wild-type cystatin F [13], using PolyJet DNA Transfection Reagent (SignaGen, Rockville, MD, USA) according to the manufacturer’s protocol. The transfected cells were left in culture for 24 h; whole cell lysates were then prepared in cell lysis buffer containing protease inhibitors, as described above, and analysed with western blot.

### 4.10. Calcein-AM Release Assay

To determine the effect of cystatin F on cytotoxicity of effector cells (TALL-104 cells and primary human SEA/SEB-stimulated CTLs), effector cells were first incubated for 4 h with 100 nM full-length or N-terminally truncated cystatin F. K-562 were used as target cells for TALL-104 cells, while SEA/SEB-pulsed Raji cells were used as targets for SEA/SEB-stimulated CTLs. Raji cells were pulsed with 1 µg/mL SEA and 1 µg/mL SEB for 1 h at 37 °C. Alternatively, to test the effect of cystatin F present in target cells on cytotoxicity, target K-562 cells were incubated for 4 h with 100 nM full-length or N-terminally truncated cystatin F. Next, cystatin F was thoroughly washed away, and the calcein-AM release assay [28] was performed, as described previously [16]. Briefly, target cells were first loaded with 15 µM calcein-AM (Sigma-Aldrich) for 30 min and then incubated with effector cells at different effector-to-target cell (E:T) ratios for 4 h. Each well of a 96-well plate contained 5000 target cells, and the following E:T ratios were prepared: 0.3:1, 0.6:1, 1.25:1, 2.5:1, and 5:1 for TALL-104 cells and 0.6:1, 1.25:1, 2.5:1, 5:1, and 10:1 for SEA/SEB-stimulated CTLs. Spontaneous release was measured by incubating target cells with complete IMDM without effector cells, while total release was measured by incubating with lysis buffer (50 mM sodium borate, 0.1% Triton X). The fluorescence of released calcein-AM was measured with a microplate reader Infinite M1000 (Tecan), and specific lysis (%) was calculated as [(test release − spontaneous release)/(total release − spontaneous release) × 100]. Lytic units (LU30/10^6^ cells) were determined using the inverse of the number of effector cells needed to lyse 30% of target cells × 100 [15].

### 4.11. Statistical Analyses

GraphPad Prism 6 (GraphPad Software, San Diego, CA, USA) was used for data analysis. Statistical significance was assessed with a *t*-test when two groups were compared or with one-way ANOVA followed by Šidák’s multiple comparisons test when more than two groups were compared. Differences were considered significant when *p* ≤ 0.05.

## 5. Conclusions

In this study, we have demonstrated that extracellular cystatin F upon internalisation decreases the cytotoxicity of cytotoxic T cells by attenuating the activities of cathepsins C and H and consequently the activities of granzymes A and B. Furthermore, we showed that cystatin F inactivates cathepsin C by direct inhibition and not by affecting processing from its inactive precursor. Similarly, cystatin F does not affect perforin processing by cathepsin L. Importantly, cystatin F secreted from target cells can also decrease T cell cytotoxicity, indicating that cystatin F can provide a protective mechanism for target cells with which they can avoid cytotoxic attack.

## Figures and Tables

**Figure 1 cancers-12-03660-f001:**
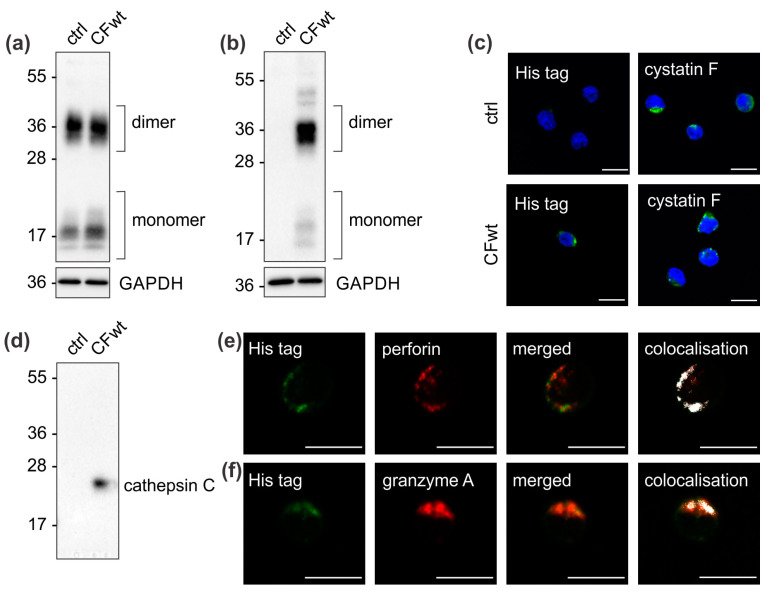
Extracellular cystatin F is internalised into TALL-104 cells. TALL-104 cells were either untreated (ctrl) or treated with 100 nM recombinant full-length cystatin F (CFwt) for 4 h. Internalisation was assessed with cell lysates and western blots using an (**a**) anti-cystatin F antibody to assess total cystatin F levels and (**b**) anti-his-tag antibody to assess internalised cystatin F levels. (**c**) Internalisation was also analysed by confocal immunofluorescence microscopy: control TALL-104 cells (first row) and CFwt-treated TALL-104 cells (second row) labelled with either anti-his-tag antibody showing recombinant cystatin F uptake (first column) or anti-cystatin F antibody showing total cystatin F levels (second column). Scale bars: 10 µm. (**d**) The interaction of internalised cystatin F with cathepsin C was demonstrated by immunoprecipitation. Cell lysates of untreated (ctrl) or CFwt-treated (CFwt) TALL-104 cells were immunoprecipitated with anti-his-tag antibody and analysed by western blot with the anti-cathepsin C antibody. (**e**,**f**) Confocal immunofluorescence microscopy demonstrating localisation of internalised cystatin F into cytotoxic granules. TALL-104 cells were incubated with CFwt and labelled with the anti-his-tag antibody (green) and either (**e**) anti-perforin (red) or (**f**) anti-granzyme A (red) antibodies. Co-localisation analysis was performed with ZEN software, and the threshold values were determined using single stained controls. Pixels with significant intensity levels of both labels are shown in white. Scale bars: 10 µm.

**Figure 2 cancers-12-03660-f002:**
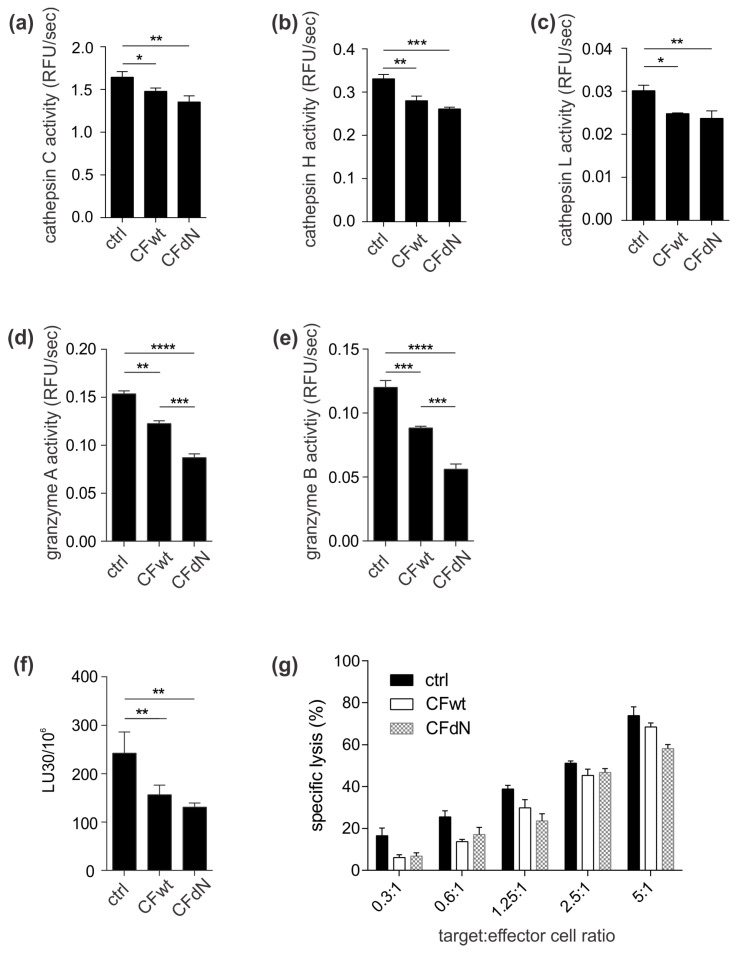
Extracellular cystatin F inhibits cathepsin C, H, and L and granzyme A and B activity as well as TALL-104 cell cytotoxicity. TALL-104 cells were either untreated (ctrl) or treated with 100 nM recombinant human full-length cystatin F (CFwt) or N-terminally truncated cystatin F (CFdN) for 4 h. The activities of (**a**) cathepsin C, (**b**) cathepsin H, (**c**) cathepsin L, (**d**) granzyme A, and (**e**) granzyme B were determined in post-nuclear cell lysates. (**f**,**g**) Cytotoxicity against K-562 target cells was measured by a 4 h calcein-AM release assay at different effector cell:target cell ratios. Lytic units (LU30/10^6^) were determined by inversing the number of TALL-104 cells needed to kill 30% of target cells × 100. Values are mean ± SD of triplicates. * *p* ≤ 0.05, ** *p* ≤ 0.01, *** *p* ≤ 0.001, and **** *p* ≤ 0.0001.

**Figure 3 cancers-12-03660-f003:**
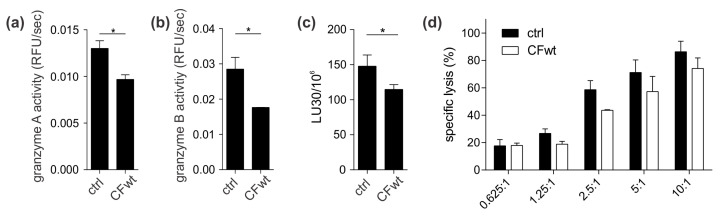
Cystatin F inhibits granzyme activity and cytotoxicity of primary human CTLs. SEA/SEB-stimulated CTLs were either untreated (ctrl) or treated with 100 nM recombinant human full-length cystatin F (CFwt) for 4 h. The activities of (**a**) granzyme A and (**b**) granzyme B were determined in post-nuclear cell lysates. (**c**,**d**) Cytotoxicity of SEA/SEB-stimulated CTLs against SEA/SEB-pulsed Raji target cells was measured by a 4 h calcein-AM release assay at different effector cell:target cell ratios. Lytic units (LU30/10^6^) were determined by inversing the number of effector cells needed to kill 30% of target cells × 100. Values are mean ± SD of duplicates (a,b) or triplicates (c,d). * *p* ≤ 0.05.

**Figure 4 cancers-12-03660-f004:**
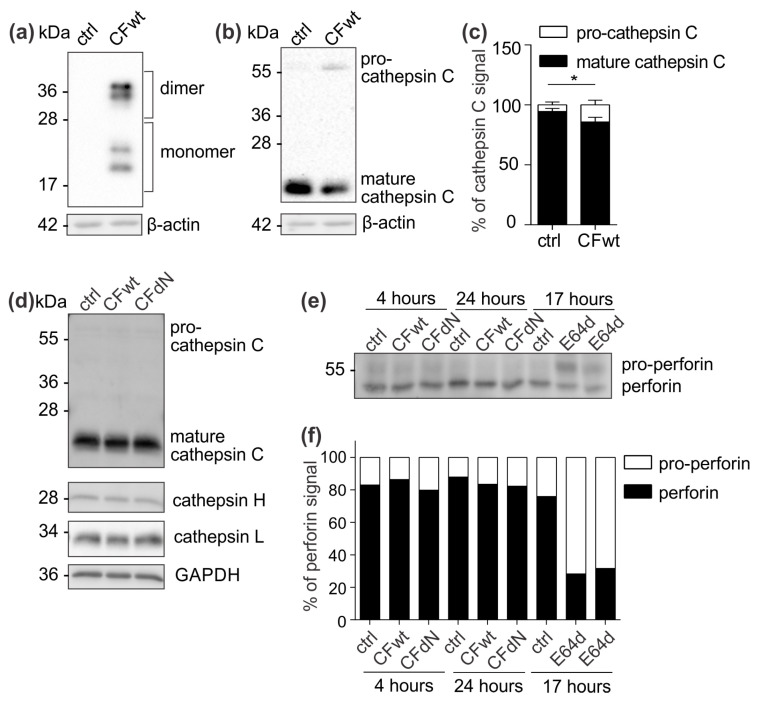
Extracellular cystatin F does not affect cathepsin C and perforin processing. (**a**–**c**) Full-length cystatin F (CFwt) was transfected into HeLa cells using the pcDNA3 vector. Transfection with an empty pcDNA3 vector served as a control (ctrl). After 24 h, whole-cell lysates were prepared and analysed by western blot. (**a**) Western blot for cystatin F confirming transfection. (**b,c**) Representative western blot and quantification for cathepsin C. Quantification was performed in Image Lab Software. Error bars represent SD between experiments. * *p* ≤ 0.05. (**d**) TALL-104 cells were treated with 100 nM recombinant human full-length cystatin F (CFwt) or N-terminally truncated cystatin F (CFdN) for 4 h, and cathepsin C, H, and L levels were determined in whole-cell lysates by western blot. GAPDH was used to confirm equal protein loading. (**e**,**f**) Western blot analysis and perforin immunodetection in cell lysates of TALL-104 cells treated with 100 nM CFwt or CFdN for 4 or 24 h and with 20 µM E64d for 17 h. Quantification of the western blot experiment shows percentages of mature perforin and pro-perforin. Quantification was performed with Image Lab Software.

**Figure 5 cancers-12-03660-f005:**
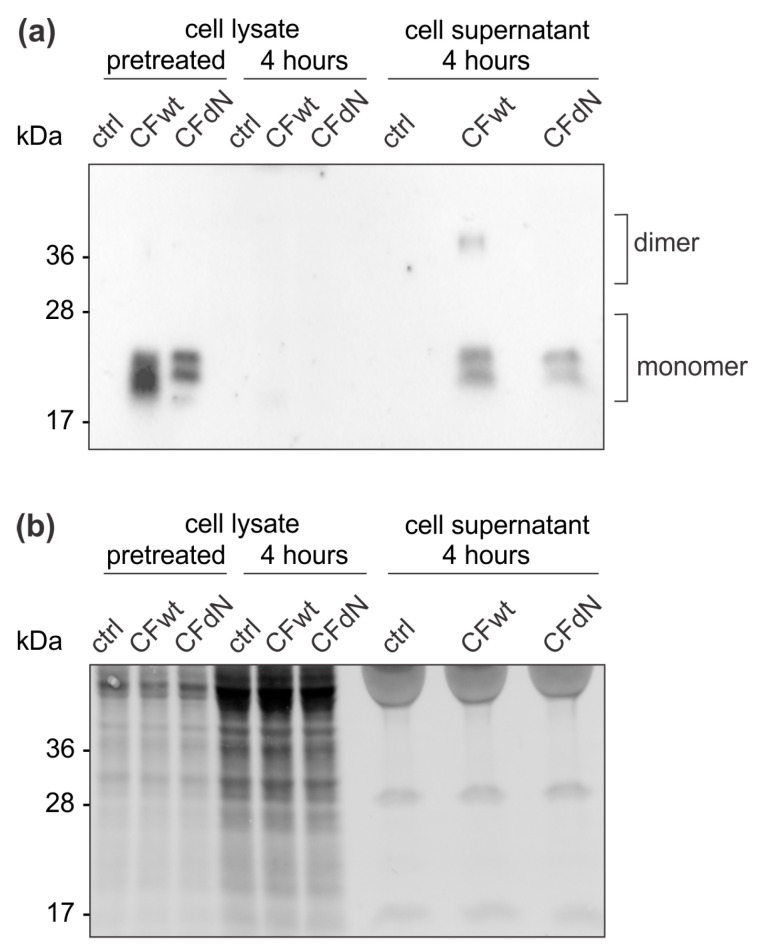
Cystatin F is internalised into target K-562 cells. K-562 cells were treated with 100 nM recombinant human full-length cystatin F (CFwt) or N-terminally truncated cystatin F (CFdN) for 4 h, then cystatin F was thoroughly washed away. Half of the cells were used to prepare cell lysate (pretreated), while the other half was resuspended in fresh media for an additional 4 h, after which cell media was collected and cell lysates were prepared. (**a**) Cell lysates and supernatants were analysed by western blot for cystatin F. (**b**) Imaging of stain-free activated proteins was used to confirm equal protein loading.

**Figure 6 cancers-12-03660-f006:**
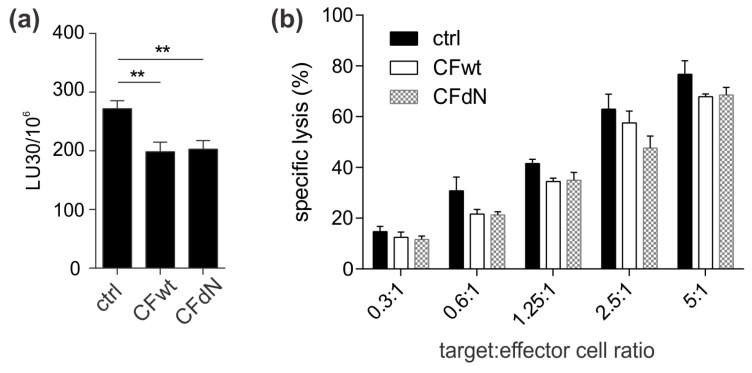
Cystatin F internalised into target K-562 cells reduces the cytotoxicity of TALL-104 cells. (**a**) Untreated (ctrl), recombinant human full-length cystatin F (CFwt)-treated, or N-terminally truncated cystatin F (CFdN)-treated K-562 cells were used as target cells, and (**b**) a calcein-AM release assay was performed at different effector cell:target cell ratios. Lytic units (LU30/10^6^) were determined by inversing the number of TALL-104 cells needed to kill 30% of target cells × 100. Values are mean ± SD of triplicates. ** *p* ≤ 0.01.

**Figure 7 cancers-12-03660-f007:**
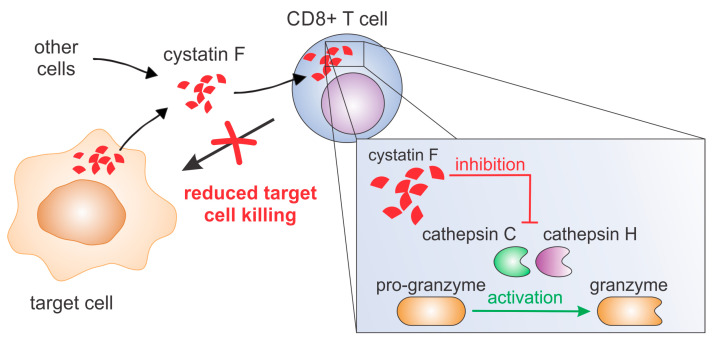
Cystatin F can be secreted from several cell types, such as monocytes, dendritic cells, mast cells or cancer cells. After internalisation into cytotoxic T cells it inhibits activities of cathepsins C and H and consequently activation of granzymes. This leads to decreased killing of target cells.

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
