# Peer review of "Extracellular Cystatin F Is Internalised by Cytotoxic T Lymphocytes and Decreases Their Cytotoxicity"

_cancers, 2020, doi:10.3390/cancers12123660_

Round 1

Reviewer 1 Report

In this manuscript, the authors report that extracellular cystatin F has a role in regulating the killing efficiency of cytotoxic T lymphocytes (CTLs). Extracellular cystatin F is internalized into the cytotoxic cell line TALL-104 leading to a decreased cathepsin C and H activity, a subsequent reduced granzyme A and B activity that a reduce their killing efficiency against K562 target. Moreover, the cystatin F release from K-562 target cells could also decrease the cytotoxicity of TALL-104 cel line.

The experiments are well designed and the results in adequacy with the figures presented. However, some questions remain and additional experiments will help to respond to them.

1- The authors highlight in the introduction that cystatin F need to monomerize to be efficient. However, in Figure 1B, the major form present is a dimer with very few monomers detected.  What is the form detected by His-Tag shown to be colocalized with perforin in TALL 104 cell line? Could the authors demonstrate that only the monomeric form is present in the cytotoxic granules?

2- the Cystatin F in its truncated form is as efficient as the wild type form in reducing the Cathepsins and Granzymes activities? How do the authors explain the absence of difference?

3- The authors present the cytotoxicity of TALL-104 against K562 using a LU3O/106 score representing the number of TALL104 required to kill 30% of K562 cells. However, it would be important to present the complete results with all the cells ratio to show that all cell ratio are affected by cystatin F.

4- How do the authors explain that, in K562 cells treated with wild-type Cystatin F, the major form detected is the monomeric form whereas in TALL-104 cells the major form is the dimeric ones?  It would be essential to confirm the experiments on the impact of Cystatin F on Cathepsins and granzyme activities with the monomeric form to be in adequacy with the model suggested by these data i.e: release of the monomeric form by K562 is taken up by TALL104 cells that will reduce their killing efficiency.

5- As K562 secretion of CF is rapid (4h)(Fig4A) it would be important to evaluate the killing of 4h pretreated cells in the presence of their supernatant or after the elimination of the supernatant ant it replacement with new medium (no more Cystatin F would be present in the supernatant) to compare the efficacy of the killing.

6- Finally as TALL-104 remains a leukemic T cell line rather than a physiologic CTL, the impact of these results would be largely increased by the addition of experiments using Ag-specific CTL clones (directed against viral or tumor antigens) to evaluate the importance of the cystatin F in their capacity to kill their specific target and validate the concept in a physiologic context.

3- The authors present the cytotoxicity of TALL-104 against K562 using a LU3O/106 score representing the number of TALL104 required to kill 30% of K562 cells. However, it would be important to present the complete results with all the cells ratio to show that all the cell ratio are affected by cystatin F.

4- How do the authors explain that in K562 cells treated with wild type Cystatin F, the major form detected is the monomeric form whereas in TALL-104 cells the major form is the dimeric ones?  It would be essential to confirm the experiments on the impact of Cystatin F on Cathepsins and granzyme activities with the monomeric form to be in adequacy with the model suggested by these data relaease of the monomeric form by K562 is taken up by TALL104 cells that will reduce their killing effcieincy

5-

Author Response

Response to Reviewer 1 Comments

In this manuscript, the authors report that extracellular cystatin F has a role in regulating the killing efficiency of cytotoxic T lymphocytes (CTLs). Extracellular cystatin F is internalized into the cytotoxic cell line TALL-104 leading to a decreased cathepsin C and H activity, a subsequent reduced granzyme A and B activity that a reduce their killing efficiency against K562 target. Moreover, the cystatin F release from K-562 target cells could also decrease the cytotoxicity of TALL-104 cel line.

The experiments are well designed and the results in adequacy with the figures presented. However, some questions remain and additional experiments will help to respond to them.

1- The authors highlight in the introduction that cystatin F need to monomerize to be efficient. However, in Figure 1B, the major form present is a dimer with very few monomers detected.  What is the form detected by His-Tag shown to be colocalized with perforin in TALL 104 cell line? Could the authors demonstrate that only the monomeric form is present in the cytotoxic granules?

Using the His-tag antibody in immunocytochemistry it is not possible to differentiate between the monomeric and dimeric cystatin F since the antibody detects both. We cannot demonstrate that only monomeric form is present in cytotoxic granules. Theoretically it would be possible to demonstrate the form of internalised cystatin F in cytotoxic granules by isolating cytotoxic granules and performing a western blot. Due to the timeframe of the revision and the amount of cells needed, the experiment is not feasible.

Perforin is co-localized with his-tag labelled both forms, that could be present in cytotoxic granules.

However, the evidence that demonstrates that a significant part of internalised cystatin F is monomeric is: (1) cystatin F needs to be monomeric and N-terminally truncated to interact with cathepsin C and interaction with cathepsin C was confirmed by His-tag pull-down (figure 1c), (2) the effect of internalised cystatin F on cathepsin activity (figure 2a-c) confirms the presence of the monomeric form, since the dimer is not active. (3) the analysis with anti-cystatin F antibodies (figure 1a) shows a similar increase of both monomeric and dimeric forms after cystatin F treatment.

Therefore, we speculate, that the amount of monomeric internalised cystatin F is not as low as the western blot for His tag might suggest. A possible explanation is that anti-his-tag antibody binds more efficiently to the dimeric form compared to the monomeric form, considering that the dimeric form has two his-tags per molecule and consequently stronger signal than the monomeric form, that has one his-tag per molecule.

2- the Cystatin F in its truncated form is as efficient as the wild type form in reducing the Cathepsins and Granzymes activities? How do the authors explain the absence of difference?

The monomeric form of intracellular cystatin F is always found in a N-terminally truncated form as published by Hamilton et al., 2008, but also our observation from several different cell lines (U937, NK92). We believe that after internalisation the wild-type form is activated by the (as of yet unknown) activating peptidase. Therefore, intracellularly they are both “truncated” and act with similar efficiency.

To make it clearer we added to the manuscript lines 247-249: “Both forms, full-length and N-terminally truncated, showed similar inhibitory activity, suggesting efficient activation of the full-length cystatin F upon internalisation (Supplementary Figure S1).”

Also we confirmed our hypothesis that internalised full-length cystatin F is indeed activated by N-terminal truncation by performing a western blot experiment with an antibody directed against the N-terminal part of cystatin F. Since no monomeric form was detected in TALL-104 cells treated with full-length cystatin F with the anti-N terminal part antibody, this indicates, that the monomeric form (detected by anti-full length cystatin F and anti-His-tag antibodies) is N-terminally truncated. We added this as Supplementary Material, Figure S1.

3- The authors present the cytotoxicity of TALL-104 against K562 using a LU3O/106 score representing the number of TALL104 required to kill 30% of K562 cells. However, it would be important to present the complete results with all the cells ratio to show that all cell ratio are affected by cystatin F.

We have included the results showing all cell ratios in the updated figures (figure panels 2g and 6b). In figure 2f,g, where cytotoxicity of TALL-104 cells pretreated with cystatin F against K562 cells was analysed, all cell ratios were affected, the effect was more pronounced at  lower E:T, for 0.3:1, 0.6:1 and 1.25 the specific lysis was app. 10% lower after treatment with wild-type or N-terminally truncated cystatin F. In figure 6, where cytotoxicity of TALL-104 cells against K-562 pretreated cells was analysed, at 0.3:1 cell ratio no difference was observed, but all other cell ratios were affected, most pronounced was the effect at 0.6:1, where specific lysis was reduced for app. 9% after treatment with wild-type or N-terminally truncated cystatin F.

4- How do the authors explain that, in K562 cells treated with wild-type Cystatin F, the major form detected is the monomeric form whereas in TALL-104 cells the major form is the dimeric ones?  It would be essential to confirm the experiments on the impact of Cystatin F on Cathepsins and granzyme activities with the monomeric form to be in adequacy with the model suggested by these data i.e: release of the monomeric form by K562 is taken up by TALL104 cells that will reduce their killing efficiency.

Possible explanations of the difference in the major form detected after internalisation are (1) that TALL-104 possess the endogenous cystatin F while K562 cells do not. After internalisation in TALL-104 cells internalised cystatin F competes with already present endogenous cystatin F for the activating peptidase and is thus converted to the monomeric form more slowly. In K562 this is not the case and the dimeric form is converted to the monomeric form faster. (2) Another explanation might be that the cell types differ in the internalisation and targeting of cystatin F.

Indeed, in the experiments of the impact of cystatin F on cathepsins’ and granzymes’ activities in addition to the wild-type form the N-terminally truncated form (solely monomeric) was always used.

Importantly, we would like to note that when K-562 cells are preloaded with the wild-type form, even if the internalised form is predominantly monomeric, both monomeric and dimeric form are found in the cell media (figure 5). Our data does not suggest that the released form by K562 is solely monomeric.

5- As K562 secretion of CF is rapid (4h)(Fig4A) it would be important to evaluate the killing of 4h pretreated cells in the presence of their supernatant or after the elimination of the supernatant ant it replacement with new medium (no more Cystatin F would be present in the supernatant) to compare the efficacy of the killing.

The experiments were performed by using K562 cells preatreated with cystatin F for 4 hours, after which the supernatant was removed and cells extensively washed. Therefore, no cystatin F was present in the media at the start of the killing (cytotoxicity assay). This was compared to untreated control cells. We did not perform experiments without changing the supernatant since we wanted to analyse the effect of CF internalised into target cells. If the supernatant with 100 nM CF would be left during the cytotoxicity assay it would not be possible to know if the observed effect is a consequence of either internalised cystatin F to K562 or of cystatin F already present in the supernatant.

6- Finally as TALL-104 remains a leukemic T cell line rather than a physiologic CTL, the impact of these results would be largely increased by the addition of experiments using Ag-specific CTL clones (directed against viral or tumor antigens) to evaluate the importance of the cystatin F in their capacity to kill their specific target and validate the concept in a physiologic context.

We would like to note that TALL-104 cells are used in the adoptive cellular therapy of cancer (brain tumours) and thus even if they might not completely reflect the physiological CTLs, understanding the role of proteolytic system in regulation of TALL-104 cytotoxicity might be beneficial for improving their therapeutic potential.

However, as suggested by the reviewer, the results of additional experiments using primary human CTLs are included in the revised manuscript.  We demonstrate that in primary human CTLs the activities of granzymes A and B are decreased after incubation with full-length cystatin F. Also, the cytotoxicity of SEA/SEB-stimulated CTLs against SEA/SEB-pulsed Raji cells is decreased after cystatin F treatment.

An additional figure (figure 3) was added and corresponding sections were updated to include the new data:

Abstract

Line 27: “... the killing efficiency of TALL-104 cells was reduced.” was changed to “the killing efficiency of TALL-104 cells as well as primary human CTLs was reduced.“

Results

Line 124:” 2.3. Cystatin F decreases granzyme A and B activity and TALL-104 cell cytotoxicity” was changed to “2.3. Cystatin F decreases granzyme A and B activity and cell cytotoxicity”

Lines 133-137 added “Lastly, to further extend our findings in a more physiological setting we tested the effect of full-length cystatin F on granzyme activity and cytotoxicity of primary human SEA/SEB-stimulated CTLs. Indeed, activities of granzymes A and B were significantly decreased (Figure 3a, b). Importantly, the ability of SEA/SEB-stimulated CTLs to kill SEA/SEB-pulsed Raji cells was also decreased (Figure 3c, d).”

Line 138-146: Figure 3 and Figure 3 legend added.

Discussion

Line 221-222: Added “A similar result was obtained using primary human CTLs.”

Line 265-268: “full-length and N-terminally truncated cystatin F caused a marked decrease in granzyme A and B activity. Furthermore, the effect of N-terminally truncated cystatin F was significantly more pronounced than that of the full-length form.” changed to: “full-length cystatin F caused a marked decrease in granzyme A and B activity in TALL-104 cells and primary human CTLs. Furthermore, the effect on granzyme A and B activity of N-terminally truncated cystatin F was significantly more pronounced than that of the full-length form in TALL-104 cells.“

Line 269-270: we added: “and decreased cytotoxicity of SEA/SEB-activated primary human CTLs against SEA/SEB-pulsed Raji target cells.”

Materials and Methods

Lines 321-332: Added “K-562 cells (CCL-243; ATCC) and Raji cells (CCL-86) (ATCC) were cultured in RPMI-1640 (Lonza) with 10% foetal bovine serum, 100 U/mL penicillin (Lonza), and 100 U/mL streptomycin (Lonza). Primary human CTLs were isolated from peripheral blood mononuclear cells (PBMC) obtained from healthy volunteers at the Blood Transfusion Centre of Slovenia, Republic of Slovenia, according to institutional guidelines. PBMCs were stimulated with 5 µg/mL staphylococcal enterotoxin A (SEA, Sigma-Aldrich) and 5 µg/mL staphylococcal enterotoxin B (SEB, Sigma-Aldrich) at a cell density of 108/mL for 1 hour and then re-suspended in RPMI (Lonza) with 10% foetal bovine serum (Gibco) and 100 IU/ml interleukin-2 (Bachem) at a cell density of 2 x 106/mL. After 5 days SEA/SEB-specific CTLs were isolated by positive selection according to manufacturer's instructions. The purity of CTLs was determined by flow cytometry using fluorescence labelled antibodies against CD3, CD4, CD8, CD56, CD16, CD19 and TCRα/β (all from Miltenyi Biotech) and was >95%.”

Lines 406-410 we added: “To determine the effect of cystatin F on cytotoxicity of effector cells (TALL-104 cells and primary human SEA/SEB-stimulated CTLs), effector cells were first incubated for 4 h with 100 nM full-length or N-terminally truncated cystatin F. K-562 were used as target cells for TALL-104 cells, while SEA/SEB-pulsed Raji cells were used as targets for SEA/SEB-stimulated CTLs. Raji cells were pulsed with 1 µg/mL SEA and 1 µg/mL SEB for 1 h at 37°C.”

Line 417 we added: “and 0.6:1, 1.25:1, 2.5:1, 5:1, and 10:1 for SEA/SEB-stimulated CTLs.”

Reviewer 2 Report

The paper of Prunk et al. describes the not further characterized uptake of extracellular cystatin F by a permanent cell line (TALL-104), which is believed to be a cytotoxic T lymphocyte model.

In this cell line, cystatin F decreased cytotoxicity, which is most likely induced by a direct decrease in cathapsin activity and the subsequent decreased activation of granzymes and perforin.

The paper is written clearly, sounds conclusive, and describes a potentially important mechanism for regulating the cytotoxicity of cytotoxic T lymphocytes.

Nevertheless, the paper does not appear to me to be publishable in its present form.

Main issues:

No significant new aspects are presented. The authors themselves cite numerous papers that covered the main aspects of the work submitted here. For example, it has already been described that extracellular cystatin F is taken up by numerous cell types and can enter vesicles of these cell types. In addition, it is known that cystatin F reduces cytotoxicity in NK and cytotoxic cells (all work from their own laboratory (9, 14, 17, 18)). The combination of experiments in the present publication may be new. However, the results were very predictable.

In order to give more weight to the findings of the present work, additional experiments are necessary. I would like to suggest at least taking a more relevant cell model and repeating the key experiments with it. The work presented was created on a permanent cell line, the physiology of which is likely to be very different from that of primary cells. Therefore, I would recommend to test significance of the results in human or mouse primary CTLs.
Another important issue is the self-citation style of the paper. 12 of 27 quotes are crossrefs from the corresponding author (I have not checked co-authors). In the DISCUSSION, 23 out of 29 mentions refer to one's own work. I would be happy if the scientific community outside the laboratory and/or network of the corresponding author could find more consideration in the discussion

Minor issues:

All figures: The panels are counted in capital letters. Lowercase letters are used in the legends. Harmonize that

L412: Supplementary Material. Not well described. There are 3 figures, no tabel, no video (at least I couldn't find it). There are no legends describing the supplementary material. Please add this.

Author Response

Response to Reviewer 2 Comments

The paper of Prunk et al. describes the not further characterized uptake of extracellular cystatin F by a permanent cell line (TALL-104), which is believed to be a cytotoxic T lymphocyte model.

In this cell line, cystatin F decreased cytotoxicity, which is most likely induced by a direct decrease in cathapsin activity and the subsequent decreased activation of granzymes and perforin.

The paper is written clearly, sounds conclusive, and describes a potentially important mechanism for regulating the cytotoxicity of cytotoxic T lymphocytes.

Nevertheless, the paper does not appear to me to be publishable in its present form.

Main issues:

No significant new aspects are presented. The authors themselves cite numerous papers that covered the main aspects of the work submitted here. For example, it has already been described that extracellular cystatin F is taken up by numerous cell types and can enter vesicles of these cell types. In addition, it is known that cystatin F reduces cytotoxicity in NK and cytotoxic cells (all work from their own laboratory (9, 14, 17, 18)). The combination of experiments in the present publication may be new. However, the results were very predictable.

Indeed, the uptake of cystatin F has been demonstrated for several cell types and in NK cells it was shown to have an impact on cell cytotoxicity. Although there are similarities in cytotoxic mechanisms between NK cells and CTLs, we cannot predict that cystatin F would have a similar effect on CTLs as on NK cells.

In order to give more weight to the findings of the present work, additional experiments are necessary. I would like to suggest at least taking a more relevant cell model and repeating the key experiments with it. The work presented was created on a permanent cell line, the physiology of which is likely to be very different from that of primary cells. Therefore, I would recommend to test significance of the results in human or mouse primary CTLs.

We would first like to note, that in addition to being a T cell model, TALL-104 cells are used in the adoptive cellular therapy of cancer (brain tumors), therefore better understanding of the role of the proteolytic system in regulation of their cytotoxicity could also be beneficial for improving their therapeutic potential. 

As suggested by the reviewer, additional experiments using primary human CTLs are included in revised manuscript to validate the concept. We demonstrate that in primary human CTLs the activities of granzymes A and B are decreased after incubation with full-length cystatin F. Also, the cytotoxicity of SEA/SEB-stimulated CTLs against SEA/SEB-pulsed Raji cells is decreased after cystatin F treatment.

An additional figure (figure 3) was added and corresponding sections were updated to include the new data:

Abstract

Line 27: “... the killing efficiency of TALL-104 cells was reduced.” was changed to “the killing efficiency of TALL-104 cells as well as primary human CTLs was reduced.“

Results

Line 124:” 2.3. Cystatin F decreases granzyme A and B activity and TALL-104 cell cytotoxicity” was changed to “2.3. Cystatin F decreases granzyme A and B activity and cell cytotoxicity”

Lines 133-137 added “Lastly, to further extend our findings in a more physiological setting we tested the effect of full-length cystatin F on granzyme activity and cytotoxicity of primary human SEA/SEB-stimulated CTLs. Indeed, activities of granzymes A and B were significantly decreased (Figure 3a, b). Importantly, the ability of SEA/SEB-stimulated CTLs to kill SEA/SEB-pulsed Raji cells was also decreased (Figure 3c, d).”

Line 138-146: Figure 3 and Figure 3 legend added.

Discussion

Line 221-222: Added “A similar result was obtained using primary human CTLs.”

Line 265-268: “full-length and N-terminally truncated cystatin F caused a marked decrease in granzyme A and B activity. Furthermore, the effect of N-terminally truncated cystatin F was significantly more pronounced than that of the full-length form.” changed to: “full-length cystatin F caused a marked decrease in granzyme A and B activity in TALL-104 cells and primary human CTLs. Furthermore, the effect on granzyme A and B activity of N-terminally truncated cystatin F was significantly more pronounced than that of the full-length form in TALL-104 cells.“

Line 269-270: we added: “and decreased cytotoxicity of SEA/SEB-activated primary human CTLs against SEA/SEB-pulsed Raji target cells.”

Materials and Methods

Lines 321-332: Added “K-562 cells (CCL-243; ATCC) and Raji cells (CCL-86) (ATCC) were cultured in RPMI-1640 (Lonza) with 10% foetal bovine serum, 100 U/mL penicillin (Lonza), and 100 U/mL streptomycin (Lonza). Primary human CTLs were isolated from peripheral blood mononuclear cells (PBMC) obtained from healthy volunteers at the Blood Transfusion Centre of Slovenia, Republic of Slovenia, according to institutional guidelines. PBMCs were stimulated with 5 µg/mL staphylococcal enterotoxin A (SEA, Sigma-Aldrich) and 5 µg/mL staphylococcal enterotoxin B (SEB, Sigma-Aldrich) at a cell density of 108/mL for 1 hour and then re-suspended in RPMI (Lonza) with 10% foetal bovine serum (Gibco) and 100 IU/ml interleukin-2 (Bachem) at a cell density of 2 x 106/mL. After 5 days SEA/SEB-specific CTLs were isolated by positive selection according to manufacturer's instructions. The purity of CTLs was determined by flow cytometry using fluorescence labelled antibodies against CD3, CD4, CD8, CD56, CD16, CD19 and TCRα/β (all from Miltenyi Biotech) and was >95%.”

Lines 406-410 we added: “To determine the effect of cystatin F on cytotoxicity of effector cells (TALL-104 cells and primary human SEA/SEB-stimulated CTLs), effector cells were first incubated for 4 h with 100 nM full-length or N-terminally truncated cystatin F. K-562 were used as target cells for TALL-104 cells, while SEA/SEB-pulsed Raji cells were used as targets for SEA/SEB-stimulated CTLs. Raji cells were pulsed with 1 µg/mL SEA and 1 µg/mL SEB for 1 h at 37°C.”

Line 417 we added: “and 0.6:1, 1.25:1, 2.5:1, 5:1, and 10:1 for SEA/SEB-stimulated CTLs.”

Another important issue is the self-citation style of the paper. 12 of 27 quotes are crossrefs from the corresponding author (I have not checked co-authors). In the DISCUSSION, 23 out of 29 mentions refer to one's own work. I would be happy if the scientific community outside the laboratory and/or network of the corresponding author could find more consideration in the discussion

By no means it is our intention to exclude any papers important to properly evaluate and discuss the results presented here. The cystatin F community is rather small and therefore, the seemingly self-citation style of the paper is a consequence of the fact that 24% of hits in PubMed when one searches for “cystatin F” include the corresponding author on the authors list. Also, in the Discussion we cite 15 papers, 6 from the laboratory/network of corresponding author. However, we are open to any suggestions for papers that would contribute to better understanding or evaluation of our results.

 Minor issues:

All figures: The panels are counted in capital letters. Lowercase letters are used in the legends. Harmonize that

All panels are now counted in lowercase letters. Similarly, in the text capital letters were also changed to lowercase letters.

L412: Supplementary Material. Not well described. There are 3 figures, no tabel, no video (at least I couldn't find it). There are no legends describing the supplementary material. Please add this.

The paper did not include any supplementary material. However, when preparing the manuscript, the line from the template describing the supplementary material was mistakenly left in the document. During the revision Figure S1 was prepared.

Line 437-438: We removed “Figure S1: title, Table S1: title, Video S1: title.” and added “Figure S1: Extracellular full-length cystatin F is activated by N-terminal truncation after internalisation into TALL-104 cells.”

Reviewer 3 Report

The study of Mateja Prunk et al entitled "Extracellular cystatin F is internalised by cytotoxic T lymphocytes and decreases their cytotoxicity" demonstrated that extracellular cystatin F upon internalisation decreases immune effector  cytotoxicity by attenuating the activities of cathepsins C and H of granzymes A and B. Similarly, cystatin F does not affect perforin processing by cathepsin L. Cystatin F secreted from target cells can also decrease cell cytotoxicity, indicating that cystatin F can provide a protective mechanism for target cells. 

This article seems to be interesting and well written. Albeit the novelty of the message is not so high,  methods, results and conclusions are quite sound. I have no major criticisms but I recommend the author to shorten very much introduction and discussion in those sections describing the general action of cystatin F in cell populations not directly involved in immune cell killing. Then, two different flow charts explaining findings of the present paper on granzyme and perforin functions upon cystatin F internalization will be welcome and help the readers to understand quickly the relevance of the overall results of the study 

Author Response

Response to Reviewer 3 Comments

The study of Mateja Prunk et al entitled "Extracellular cystatin F is internalised by cytotoxic T lymphocytes and decreases their cytotoxicity" demonstrated that extracellular cystatin F upon internalisation decreases immune effector cytotoxicity by attenuating the activities of cathepsins C and H of granzymes A and B. Similarly, cystatin F does not affect perforin processing by cathepsin L. Cystatin F secreted from target cells can also decrease cell cytotoxicity, indicating that cystatin F can provide a protective mechanism for target cells. 

This article seems to be interesting and well written. Albeit the novelty of the message is not so high, methods, results and conclusions are quite sound. I have no major criticisms but I recommend the author to shorten very much introduction and discussion in those sections describing the general action of cystatin F in cell populations not directly involved in immune cell killing.

We have included the detailed description of cystatin F actions in the introduction and discussion sections so that the subject can be followed more easily by a non-specialist reader. As suggested by the reviewer, we have shortened the introduction and discussion sections to make them more concise. The changes were made in the following lines 43, 48, 61, 204, 206, 209, 212, 217 and 256.

Then, two different flow charts explaining findings of the present paper on granzyme and perforin functions upon cystatin F internalization will be welcome and help the readers to understand quickly the relevance of the overall results of the study 

As suggested by the reviewer, a schematic representation of the presented results is now included as Figure 7.

Round 2

Reviewer 1 Report

The authors have responded to the major criticisms by performing new experiments in primary CTL and modified the manuscript to better explain the results obtained.

Reviewer 2 Report

All questions were answered convincingly by the authors.